# Contrasting Management and Fates of Two Sister Lakes: Great Salt Lake (USA) and Lake Urmia (Iran)

Wayne A. Wurtsbaugh [1,*] and Somayeh Sima [2]

1   Department of Watershed Sciences and the Ecology Center, Utah State University, Logan, UT 84322, USA
2   Faculty of Civil & Environmental Engineering, Tarbiat Modares University, Tehran 14115-111, Iran
*   Correspondence: wayne.wurtsbaugh@usu.edu; Tel.: +1-435-760-7825

**Abstract:** Many saline lakes throughout the world are shrinking due to overexploitation of water in their drainage basins. Among them are two of the world's largest saline lakes, the U.S.A.'s Great Salt Lake, and Iran's Lake Urmia. Here we provide a comparative analysis of the desiccation of these two lakes that provides insights on management decisions that may help save them and that are relevant to saline lake management worldwide. Great Salt Lake and Lake Urmia were once remarkably similar in size, depth, salinity, and geographic setting. High rates of population growth in both basins have fueled a demand for irrigated agriculture and other uses. In the Great Salt Lake basin, this development began in the late 1800's and is continuing. The lake's volume has decreased by 67%, with 75% of the loss driven by water development and 25% by a millennial drought which may portend the start of global climate change impacts. This has greatly increased salinities to 180 g·L$^{-1}$ stressing the invertebrates in the lake on which birds depend. Only 1% of people in the basin are employed in agriculture; thus, reducing the demand for irrigation development. Population densities in the Urmia basin are double those of the Great Salt Lake basin, and 28% of people are employed in agriculture. These demographics have led to a rapid increase in reservoir construction since 2000 and the subsequent loss of 87% of Lake Urmia's volume. The water development of Lake Urmia was later, but much faster than that of Great Salt Lake, causing Urmia's salinity to increase from 190 to over 350 g·L$^{-1}$ in just 20 years, with subsequent severe ecological decline. Dust storms from the exposed lakebeds of both systems threaten the health of the surrounding populations. To save these lakes and others will require: (1) transparent and collaborative involvement with local interest groups; (2) shifts away from an agricultural-based economy to one based on manufacturing and services; (3) consideration of the diverse ecosystem services of the lakes including mineral extraction, recreation, bird habitats in surrounding wetlands, and dust control.

**Keywords:** saline; agriculture; lakes; management; population; ecosystem; economy; ecology; agriculture

## 1. Introduction

Many saline lakes worldwide are in trouble. In their endorheic basins, lake areas decrease so that evaporative losses balance the decreasing inflows. In some cases, the decreased inflows are the result of climate change [1], but in most cases, demands for water to support ever-increasing populations are the major factor impacting these terminal lakes [2,3]. However, increasing temperatures and changes in precipitation patterns due to anthropogenic climate change are an ominous threat, both because of direct decreases in a runoff, and indirect effects due to increased irrigation of croplands in a warmer climate.

The shrinkage of saline lakes causes many direct and indirect economic and environmental costs [4,5]. Dominant industries in hypersaline lakes are mineral extraction from salt ponds, the harvest of *Artemia* cysts for aquaculture, and in hyposaline lakes, fish production can be important. These economic values are often underappreciated, but the value of the U.S.'s Great Salt Lake is estimated at $1.7 billion, and the commercial fishery that once existed in the greater Aral Sea supported 60,000 jobs [6]. Environmental costs of saline

lake desiccation are also high. Terminal lakes accumulate minerals and nutrients and, thus, they often produce high quantities of invertebrates. In hypersaline lakes lacking fish, the high production of invertebrates can be channeled to birds, making these lakes important magnets for their nesting and feeding [7]. When lakes desiccate, and their salinity rises above 180–200 g/L, salt-tolerant *Artemia* and brine flies (*Ephydra* spp.) become stressed and densities decline; thus, limiting prey for birds [8–10]. In hyposaline systems, increasing salinities and lake areas have decimated important sport and commercial fisheries (e.g. Salton Sea and Aral Sea; [11,12]). Shrinking salt lakes also limit access to boating, swimming, and other recreational activities [13,14]. Finally, the dried lakebeds of desiccated lakes often produce dust harmful to human health (e.g. Aral Sea; [15]). For example, the small (285 km$^2$) dried lakebed of Owens Lake (California) produces the most dust pollution in the U.S.A, causing health problems for the surrounding population and more than US$3.6 billion will be spent over 25 years to remediate this problem [16].

These economic and environmental costs of lake desiccation are being imposed on two of the largest saline lakes in the world, the U.S.'s Great Salt Lake and Iran's Lake Urmia. In 2022 Great Salt Lake reached its lowest level in recorded history and Lake Urmia's area has been reduced by 69% compared to its maximum extent, and Iranians are struggling to recover it [17]. Here, we provide a comparative analysis of the desiccation of the two lakes that provides insights on management decisions that may help save them and that are relevant to saline lake management worldwide.

## 2. Similarities and Differences between Lake Urmia and Great Salt Lake

### 2.1. Geographical and Physical Factors

Great Salt Lake in the western United States and Lake Urmia in western Iran are remarkably similar in many characteristics (Figure 1; Table 1), and both are threatened by agricultural water diversions and pollution from nearby cities. Although the sizes of both lakes fluctuate due to climatic cycles and water diversions, the unimpacted (natural) areas of both were similar (Great Salt Lake 4980 km$^2$; Urmia 4630 km$^2$), and they lie at similar elevations (Great Salt 1282 m; Urmia 1276 m). The mean and maximum depths of Great Salt Lake at its mean natural elevation (1282 m) were 5.7 m and 12.3 m, respectively, whereas those for Lake Urmia at its natural elevation were 3.9 and 10.8 m. Both lakes have been divided by causeways: Great Salt Lake by a railroad causeway and Lake Urmia by an automobile causeway. Passage of water between the two major parts of Great Salt Lake is now restricted by an 82-m breach located in shallow water and, until recently, by two 5-m wide culverts. Consequently, a lack of extensive water exchange allows major differences in salinity in north and south basins, which in turn allow distinct biota to grow (note the color differences, Figure 1). A much larger 1250-m gap in Lake Urmia's causeway apparently allows more mixing between the basins, so that differences in salinity are minimal [17,18]. However, both lakes are hypersaline and in their natural state the dominant ions were Na$^+$ and Cl$^-$.

There are major geographical similarities and differences as well. The watersheds of both lakes pass through three different States or Provinces, thus, complicating water management issues. The catchments of the two lakes are similar: 55,700 km$^2$ for Great Salt Lake and 52,000 km$^2$ for Lake Urmia [19]. A major difference is that Lake Urmia lies at a lower latitude (37° N) than Great Salt Lake (41° N), but seasonality is similar for both, with hot summers and cold winters. The mean annual temperature for Great Salt Lake is 11.5 °C, and the freeze-free period lasts for an average of 167 days. For Lake Urmia, the mean annual temperature is 11.4 but with a longer freeze-free period of 232 days. The longer freeze-free period for Urmia means that irrigated farming is more prolonged than in the Great Salt Lake watershed. Mean precipitation in Great Salt Lake catchment is considerably higher (545 mm·yr$^{-1}$) than reported for Lake Urmia (350 mm·yr$^{-1}$) [17,20].

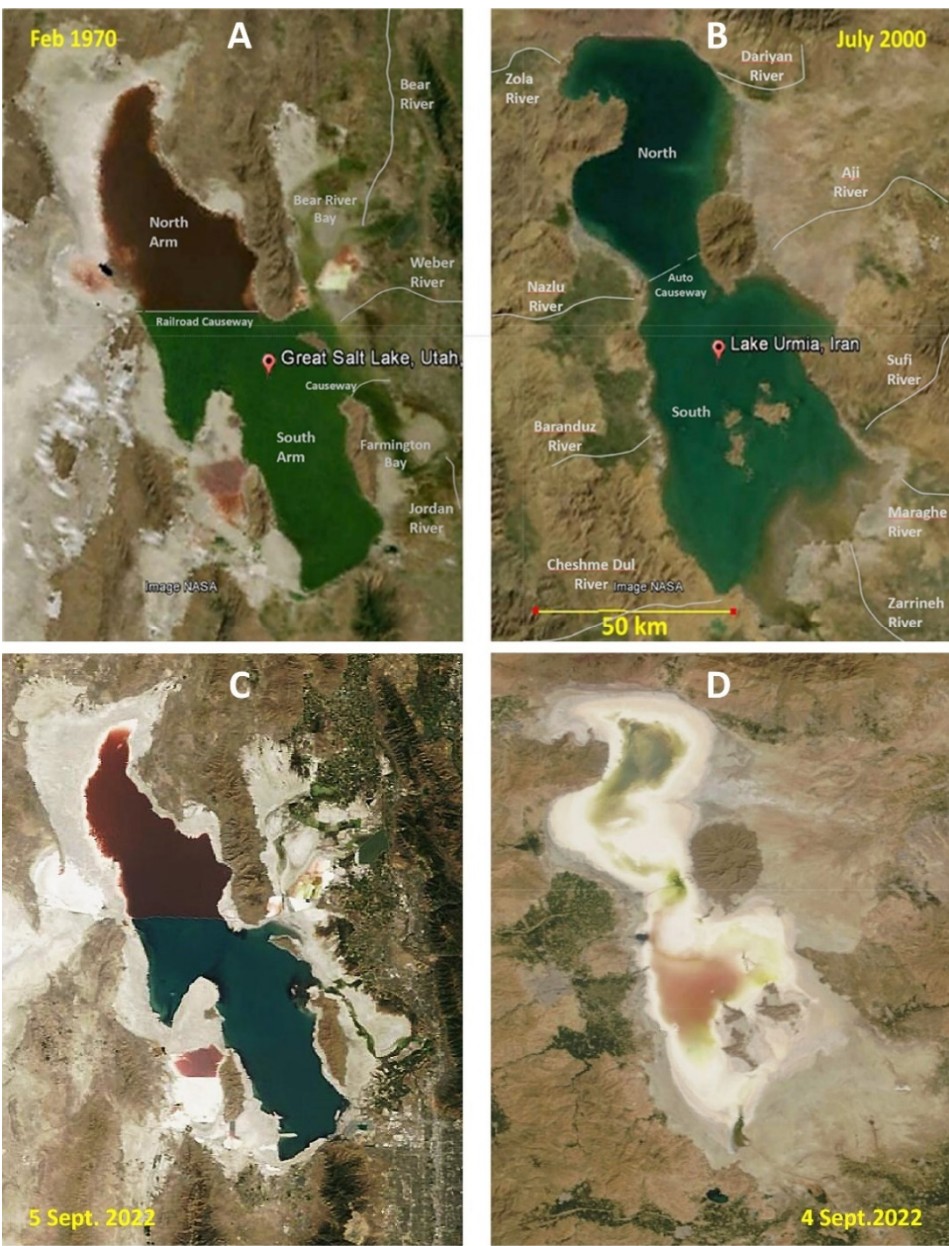

**Figure 1.** (**A**,**B**) Satellite imagery of Great Salt Lake (**left**) and Lake Urmia when both were moderately full. Note that transportation causeways divide each lake into northern and southern sections. In Great Salt Lake this results in a much higher salinity in the north arm, and the microbes in the salt-saturated water turn it red. The Bear River is now the major source of freshwater for Great Salt Lake, and the Zarrineh River in the south provides the majority of inflow to Lake Urmia. (**C**,**D**) Recent images of Great Salt Lake and Lake Urmia showing the dramatic desiccation, particularly of Lake Urmia.

There are important similarities and differences in the human geography of the two lake basins. Both have high populations near the lakes. Great Salt Lake is bordered on its eastern and southern shores by a metropolitan area with 2.5 million people, whereas the city of Urmia, with 1.2 million residents, is situated 13 km west of the lake and Tabriz, with 1.6 million residents, is located 95 km east. However, a total of 5.35 million people live in Lake Urmia's watershed [18], nearly double that in the Great Salt Lake watershed (2.9 million; 2020 census). Irrigated agriculture is more dominant in the Urmia basin, covering 6000 km$^2$ (12%) whereas it represents only 1600 km$^2$ (3%) n the Great Salt Lake basin. Agriculture is also a small employer of the population in the Great Salt Lake basin

(1%) in comparison with 28% in the Urmia basin (data from Statistical Center of Iran and [22]).

**Table 1.** Characteristics of Great Salt Lake and Lake Urmia and their basins. A * indicates the size at their natural elevation prior to major water developments. For Lake Urmia, this was calculated based on mean lake elevations from 1910–1995, although we recognize that irrigation practices were in place in the late 1800s (Günther 1899) [21] that may have decreased lake size. For Great Salt Lake, the natural elevation was modeled based on water depletions and measured elevations from 1847–2007 [3] (Wurtsbaugh et al., 2017).

| Characteristic | Great Salt Lake | Lake Urmia |
| --- | --- | --- |
| Elevation * (m) | 1282.3 | 1274.9 |
| Natural Lake Area * ($km^2$) | 4980 | 4630 |
| Current Lake Area (4 September 2022, $km^2$) | 2303 (46%) | 1427 (31%) |
| Lake Volume * ($km^3$) | 28.6 | 18.2 |
| Mean Depth * (m) | 5.7 | 3.9 |
| Watershed Area ($km^2$) | 55,700 | 52,000 |
| Irrigated agriculture ($km^2$) | 1600 | 6000 |
| Population in watershed (Million) | 2.83 | 5.35 |

Populations in both Iran and Utah are growing very rapidly with annual increases of nearly 2% (Figure 2). Human densities in the lake basins are also increasing rapidly, but densities are two times higher in the Lake Urmia basin than in the Great Salt Lake basin (Figure 2B). Population growth is, thus, a major factor driving increased use of water, particularly for agriculture.

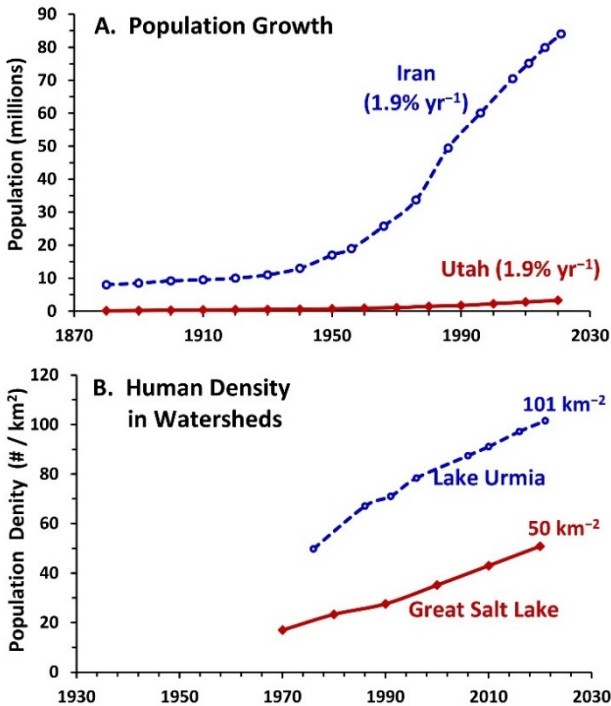

**Figure 2.** (**A**) Total populations of Iran and Utah, showing rapid increases. The respective areas of Iran and Utah are 1,648,200 and 219,900 $km^2$. (**B**) Population densities in the Lake Urmia and Great Salt Lake watersheds since the 1970s.

### 2.2. Desiccation of the Lakes

Both Great Salt Lake and Lake Urmia have suffered major decreases in lake elevation and volume (Figure 3), primarily as a result of water depletions for agriculture and other

uses. Extensive water development in the Urmia basin occurred later, but much more rapidly than in the Great Salt Lake basin, so the decline in Lake Urmia has been precipitous and more easily recognized (Figures 3 and 4). Highly variable climatic conditions have produced high fluctuations in lake levels. During a wet cycle in the mid-1980s, Great Salt Lake reached its highest level and flooding was severe on infrastructure that had encroached on the lake's shore. Nevertheless, the overall trend in lake elevation has been significantly downward and a recent 20-year drought has exacerbated the situation. In September 2022, the lake reached its lowest recorded level of 1276.8 m (4189.1 ft.), with lower levels expected by the end of 2022. The decline has exposed over 54% of the lakebed. However, in the highly productive, less saline, estuaries of Farmington and Bear River Bays on the east side of the lake, more than 80% has been exposed. These estuaries are extremely important bird habitats [23,24].

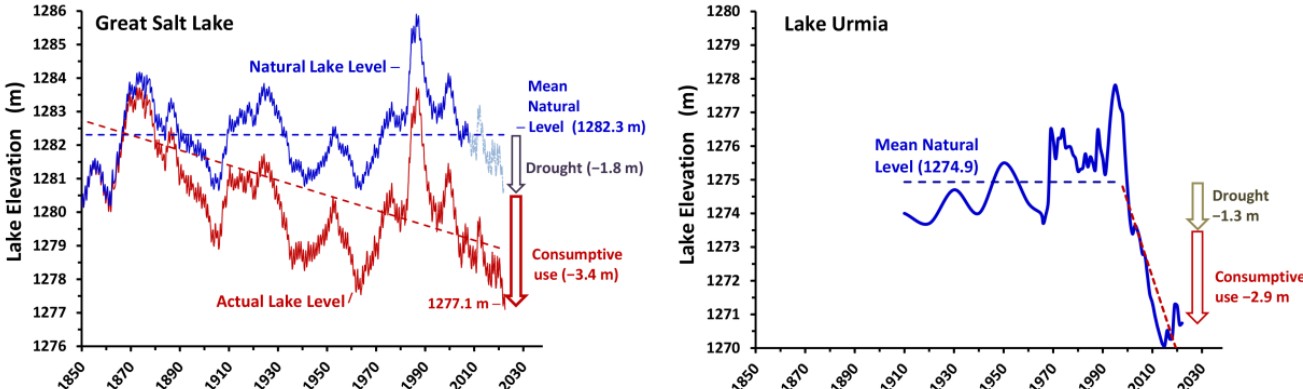

**Figure 3. Left**. Elevation of Great Salt Lake (red) and the modeled 'natural' level of the lake if there had been no consumptive use (blue). The light blue line represents an extrapolation of the model beyond 2007. **Right**. Elevations of Lake Urmia showing the mean "natural level" from 1910–1995, and the rapid decline since then. The dashed red line for Lake Urmia shows the regression for the period of intensive dam building (1998–2020).

The decline in Great Salt Lake's level is largely the result of water diversions for agriculture and other uses in the basin [3,25]. Fifty percent of water depletion occurred before 1900 in the GSL basin [3] and major dam construction began in the 1920s (Figure 4). A model using water depletions from various land uses (Figure 5) indicated that water diversions reduced river inflows by 39% and lowered the lake by 3.4 m (11 feet) from its natural level [3] (Figure 3). As a result of the hypsographic shape of the basin [26], this 3.4 m decrease in elevation represents approximately a 50% decrease in the volume of the lake. However, this decline has been ameliorated because of two reasons. First, irrigated agricultural land in the basin is being converted to urban use, and water loss is not as great in urban areas as it is with irrigated land. From 1949 to 2003, irrigated land in Great Salt Lake basin has declined from 1900 km$^2$ to 1600 km$^2$—a 16% reduction. Secondly, about 0.2 km$^3 \cdot$yr$^{-1}$ of water from the Colorado River Basin is diverted into the Salt Lake Valley, partially making up for agricultural water withdrawals. However, this inter-basin water transfer only supplies about 1.5% of the lake's water, has taken over 50 years to complete, and when finished, will have cost approximately $3 billion US. The long delays, environmental costs [27], and high monetary and cultural costs inevitable in inter-basin water transfers must be taken into account when planning for the restoration of any lake [28].

Despite the fact that Great Salt Lake has reached its lowest level ever and exposed over 50% of the lakebed, planning is underway to utilize an additional 20% of the flow of the Bear River, the major tributary feeding Great Salt Lake [29]. If this plan is implemented, additional loss in the volume of Great Salt Lake is expected. Mohammed and Tarboton [30] conducted a sensitivity analysis of the lake to changes in runoff and evaporation and

estimated that a 25% reduction in all river flows would reduce the elevation of Great Salt Lake by another 0.7 m (2.3 ft).

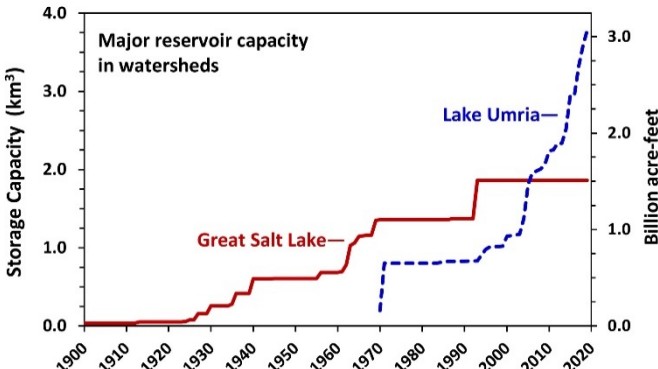

**Figure 4.** Increase in major reservoir storage capacity in the Great Salt Lake and Lake Urmia watersheds. Data for Great Salt Lake and Lake Urmia watersheds were derived, respectively, from [31] and the Iran Water Resources Management Company.

Climate change may also impact lake levels, but the uncertainty in local climate models makes predictions difficult [1,32]. The 1.4 °C increase in local temperatures has increased lake evaporation and led to an estimated decrease in lake level of 0.1 m, but this is small relative to the decrease due to water development in the basin [33]. However, increasing temperatures in the basin will also likely decrease runoff, because of increases in evapotranspiration and earlier snowmelt [34]. Consequently, the potential for increased water development, increased lake evaporation at higher temperatures, and decreased runoff pose serious threats to the long-term elevation and salinity of Great Salt Lake.

Two main reasons have been identified as the root causes of Lake Urmia desiccation: a 20% decline in the total renewable water resources due to a change in the meteorological condition of the basin, and a 50% decrease in river inflows to the lake since 1995. The decreased river inflow is attributed to: (1) a doubling of irrigated areas since the 1980s (from 3000 ha to ~6000 ha) [35]; (2) an increase in the number of wells (currently 89,000 wells) [36]; and (3) dam construction. A review of the literature by Parsinejad et al. [17] suggests that approximately 68% of the decline in lake level has been due to water depletions and 32% to climate change. Like Great Salt Lake, precipitation has not declined significantly in the Urmia Basin ($p = 0.26$; Parsinejad et al., 2021) (Figure A1), but long-term air temperature records in the Lake Urmia basin do show a slightly increasing trend (0.05 °C·yr$^{-1}$) [35,37]. Currently, 74% of the basin's 7140 million m$^3$·yr$^{-1}$ of freshwater resource is applied for human use, which falls in the high water stress level category defined by the United Nation's sustainable development goal [38]. Agriculture consumes 89% of the freshwater used in the basin (Figure 5).

The management goal is to reduce this to 40% by 2026, but progress has been slow to implement this change. Additional savings will be difficult because most projections of climate models show a rise in the air temperature and a decline in the precipitation of the basin by 2050 (e.g. [39,40]) If this occurs, evapotranspiration, and, thus, irrigation water consumption will increase. Simultaneously, population growth will cause an increase in the food demand and, thus, irrigated areas and exacerbate increases in water consumption in the Urmia basin.

Based on gauge data prior to major dam construction, the minimum and maximum elevations of Lake Urmia were 1273.5 in 1966 and 1278.4 m in 1995, respectively (Figure 3). Since 1995, a downward trend of lake level has been recorded as a result of substantial water storage projects and agricultural development. Lake Urmia's basin major dam construction began in 1998 and it has increased water storage capacity in the Lake Urmia basin to more than double that of storage in the Great Salt Lake Basin (Figure 4). Between 2002 and 2013, the total inflow to the lake declined at a rate of 303 million m$^3$·yr$^{-1}$ and caused a

$0.35 \ \mathrm{m \cdot yr^{-1}}$ decrease in lake level. At water levels between 1272 and 1277 m, a 1 m drop in the lake level causes a 5 km$^3$ reduction in the lake volume [41]. Moreover, at water levels below 1272 m, the exposure of the lakebed increases significantly (>300 km$^2$) with the potential to create dust storms [14,17]. In the past decade Lake Urmia's level at the end of the water year has been always below this threshold level (Figure 3). Following the model used for Great Salt Lake, Sima et al. [10] produced a matrix identifying how different beneficial uses of Lake Urmia could be met at different lake elevations.

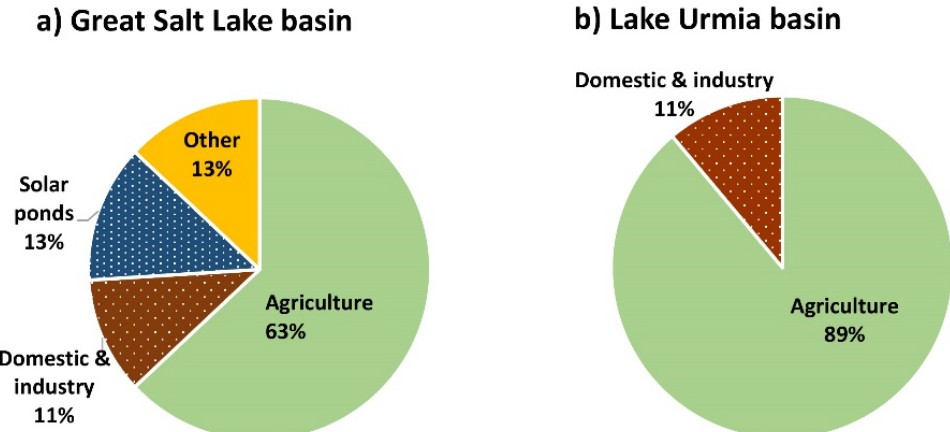

**Figure 5.** Current percentages of water use in the Great Salt Lake and Lake Urmia basins by different sectors. "Other" includes evaporative loss from diked wetlands and reservoirs. Data: Wurtsbaugh et al. (2017) [3] and data extracted from Lake Urmia's Water Allocation Document (2017), Iran Ministry of Energy.

Between 2013 and 2019 the Urmia Lake Recovery Project (ULRP) and high precipitation years helped stabilize the lake level and prevented further decrease [42]. However, in September 2022 the lake level declined to an elevation of 1270.0 m (volume = 1.5 km$^3$; 8% of the natural volume which was close to the lake elevation at the onset of ULRP in October 2013 (Figure 3). This has been interpreted as a failure of the restoration efforts and raised debates. There are also concerns about the complete drying of the southern part of Lake Urmia by the end of the current water year (September 2022; Figure 1).

*2.3. Salinity Changes*

With desiccation, increasing salinities are a concern for both lakes primarily because brine shrimp and brine flies cannot tolerate extreme conditions. In Great Salt Lake salinities have sometimes reached saturation levels of NaCl [31], but with the completion of the solid-fill railway causeway in 1959 the north and south arms began to diverge (Figure 6). All of the major rivers flow into the south arm, whereas the north arm receives most of its water from the south arm through breaches in the causeway, and by direct precipitation. As a consequence of the division, salinities in the south arm declined steadily, reaching levels near 60 $\mathrm{g \cdot L^{-1}}$ during the 1980s when extreme precipitation in the basin raised the lake level to near-record levels. Since then, salinities in the south arm have generally ranged from 90–180 $\mathrm{g \cdot L^{-1}}$, but with a strong increasing trend during the recent drought.

The south arm of Great Salt Lake also develops a deep brine layer (monimolimnion) when the lake is greater than 6 m deep, the approximate thickness of the mixed layer. The high-salinity water maintaining the deep brine layer enters from the north arm as an underflow through the breach and causeway fill material. This layer accumulates sedimenting organic matter, becomes anoxic with reducing conditions, and high concentrations of hydrogen sulfide and methyl mercury are produced [43–45], making it uninhabitable for brine shrimp and brine flies [46]. An estimated 40% of the toxic deep brine layer is entrained into the upper mixed layer each year [45].

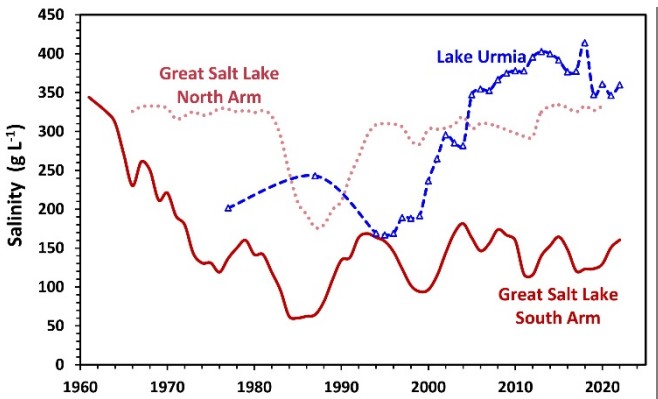

**Figure 6.** Salinities in the north and south arm of Great Salt Lake and in Lake Urmia. Salinities in Great Salt Lake were measured at open lake stations at depths of 0.5–3 m and averaged. Salinities for Lake Urmia were normally measured with surface samples collected at the breach between the north and south arms.

Salinities in the north arm of Great Salt Lake have usually been near saturation (>300 g/L), except in the mid-1980s period when lake levels rose appreciably and diluted the basin. The saturated water has led to halite deposition in the sediments, but the magnitude of this has not been measured [47]. The high salinities in the north arm are due, in part, to the migration of salts from the south into the north where they precipitate [47]. The high salinities in the north arm provide a concentrated brine for the potash extraction industry, whereas the lowered concentrations in the south arm are a concern for the magnesium extraction industry there. Managers have some capacity to regulate brine flow from the north to the south using underwater berms near the breach, and these are being used to try to fulfill the competing needs of the two primary mineral extraction companies, and the brine shrimp harvesting industry located on the south arm. However, the berms lack precise control and adjustable mechanical gates will be needed to better regulate the north and south flow of the brines [48].

Although data are limited, salinities in Lake Urmia generally ranged from 170–240 $g \cdot L^{-1}$ prior to major reservoir development and the desiccation of the lake (Figure 6). Since 2000, salinities have risen steadily and are now 300–400 $g \cdot L^{-1}$ NaCl (halite) precipitation in the lake is extensive and has modified the lake's morphometry. Mohammadi et al. [49] estimated that since the construction of major dams, the sedimentation/salt accumulation rate has increased from 0.3 $mm \cdot y^{-1}$ to 66 $mm \cdot y^{-1}$ generating a salt crust with a maximum thickness of 2.95 m. With the precipitation of halite, the lake's salts are now balanced between Mg, $SO_4$, Na, and Cl [14]. The increasing concentrations of Mg and $SO_4$ present opportunities for mineral extraction from the lake (see below).

### 2.4. Biological Factors

Both lakes had extensive natural wetlands. At Great Salt Lake, the three major tributary rivers now flow through diked wildlife refuges with freshwater wetlands that cover 226 $km^2$ [50,51]. Many of these were developed from the 1920s–1950s to provide nesting and resting habitats for shorebirds, migratory waterfowl, and other birds. Water from these wetlands, and from small creeks flowing across playas around the margin of the lake, add additional wetland area. The importance of these wetlands for birds led the lake to be added to the National Hemispheric Shorebird Reserve in 1991. At Lake Urmia, wetlands are estimated to now cover 260 $km^2$ (Personal communication, West Azerbaijan Bureau of Environment), although desiccation, channelization, and eutrophication are degrading many of them [52]. The lake's former importance for birds led to it being designated as one of the first RAMSAR Wetlands of International Importance in 1975 [53].

Both lakes have had abundant populations of brine shrimp (*Artemia* spp.) [54–57]. At Great Salt Lake, the resting eggs (cysts) of the *Artemia franciscana* are harvested and shipped

worldwide to support prawn and finfish aquaculture. This industry is valued at US $70 million in today's dollars [4]. In Great Salt Lake, *Artemia* are abundant in the southern basin, but the saturated salts in the basin north of the railway causeway allow only limited brine shrimp reproduction, and densities there are low [8]. As the lake continues to fall, salinities in even the southern basin are rising and beginning to stress the *Artemia* and reduce reproduction [8,56,57]. Managers are now addressing how to decrease salt transport from the north basin through the causeway breach into the southern basin of the lake [58]. The *A. urmiana* in Lake Urmia appear to be more salt-tolerant than the *A. franciscana* in Great Salt Lake, but populations there have not been carefully monitored. However, a compilation of data from various sources suggests that by 2013 the increasing salinity in Lake Urmia had reduced densities to less than 35% of those before desiccation began (Sima et al., 2021). Dahesht, et al. [59] report that there was an artisanal industry of cyst collection from the lakeshore.

Both lakes have populations of brine flies (*Ephydra* spp.) [46,54] and at Great Salt Lake, these are most abundant on the solid structure of microbialites (stromatolites, etc.) that ring around much of the southern basin [7,60]. Little is known about the abundance and distribution of brine flies in Lake Urmia.

Migratory birds use both lakes and their wetlands extensively [61,62]. Annually, Great Salt Lake and its wetlands host over 10 million birds, represented by 338 species. In Great Salt Lake the freshwater and brackish wetlands attract a greater diversity of birds than the open saline lake [23,63]. Increases in spring and fall water flow into the lake's eastern bays increase the densities of many bird species [24]. The diets of many of the birds in Great Salt Lake have been studied, and the brine flies, freshwater invertebrates, and *Artemia* are all important to different species [7,23,64,65].

Overall, 212 species of birds have been identified in Lake Urmia's ecological zones (lake body, islands, wetlands, and bays) [66,67]. However, continuous counts of birds and abundances have not been reported, with the exception of annual censuses of dominant birds such as the Greater flamingo (*Phoenicopterus roseus*), which indicate that densities declined to near zero as salinities in the lake climbed to $>340$ g·L$^{-1}$ [14]. The diets of birds that inhabited Lake Urmia are unknown.

## 3. Discussion

### 3.1. Lake Desiccation

Great Salt Lake and Lake Urmia were once remarkably similar, and both have been desiccated to some extent, but Lake Urmia has experienced more intense desiccation and has lost most of its ecological function and beneficial uses [14], whereas Great Salt Lake has retained much of its functionality but continues to be threatened by declining lake levels and increasing salinities [68]. Although the total decline in lake level has been less for Lake Urmia than for Great Salt Lake, the impact has been more severe for Lake Urmia because its mean depth was 1.6 m less than that of Great Salt Lake. Additionally, the rate of lake-level decline (Figure 3) and the increase in salinity (Figure 6) have been much higher in Lake Urmia than in Great Salt Lake. The review by Parsinejad et al. [17] suggests that 62% of the decline in Lake Urmia has been due to water development projects, and 38% due to climate changes, and the importance of the former is supported by the coincidence of rapid lake level decline with the increase in reservoir storage capacity (c.f. Figures 3 and 4). For Great Salt Lake, water development is also the dominant factor driving down lake levels [3,33], but long-term drought or climate change is likely gaining in importance [34,69]. Although climate change is a very serious long-term threat, controlling the over-consumption of water is a more tractable solution for saving saline lakes [1,70] that can be addressed at the national or state/provincial level of administration.

In many saline lake basins, water development has been gradual, and thus the cause of declining lake levels has not been obvious (e.g., Great Salt Lake, Lake Abert [71], Oregon, USA), particularly when climatic cycles raise and lower the lakes dramatically, making trends difficult to observe. Careful modeling was necessary to understand that water

development was, indeed, the major factor desiccating these two lakes [3,71]. For Lake Urmia and the Aral Sea [72], the rapid declines in lake level coincided with major dam construction and irrigation projects, so it was easy to attribute the lake's demise to water development. Water balance models are, thus, critical for assessing the reasons for the decline of any saline lake [3].

The underlying problem threatening Great Salt Lake, Lake Urmia, and most saline lakes is unsustainable population growth that increases demand for irrigated agriculture and urban water supplies. Many endorheic basins with saline lakes have warm, dry summers that support intensive agriculture if fresh water is available, but the development of that water in endorheic basins usually diminishes saline lakes. This problem has been more severe in the Urmia basin than in Great Salt Lake's because population density is two times higher and because of a governmental plan for food production independence, in part driven by economic sanctions imposed by the U.S. and other western countries [73]. Madavi Madani [74] argues that water mismanagement in Iran is also a serious contributor to shortages and that the economic sanctions have intensified it and have had negative consequences for Iran's environment [75]. The Lake Urmia basin, and indeed, most of Iran, also has limited agricultural land, with most crop production occurring in irrigated river valleys. In contrast, the U.S. has extensive agricultural lands that can supply food to Utah; in fact, most counties in the Great Salt Lake basin import more than 85% of their food from other states [76]. The current demand for water development in Utah is driven not by agricultural needs but perceived needs of a growing urban population [77,78].

*3.2. Economic and Cultural Factors Influencing the Lakes*

Economic and cultural interests in the two lakes have also influenced their protection and management. Great Salt Lake's US$1.7 billion economic value derives primarily from mineral extraction (87%), recreation (9%), and *Artemia* harvest (4%). Commercial mineral extraction began in the 1860s and the principal products produced now are magnesium and potash fertilizers [4,79]. Diking and protection of the surrounding wetlands began around 1900 for sport hunting of waterfowl and in 1929, the 299 km$^2$ National Bear River Migratory Bird Refuge was established in the wetlands. Swimming and boating became popular in the late 1800s and several resorts were established within easy reach of metropolitan Salt Lake City. The intensive harvest of *Artemia* cysts for aquaculture production worldwide began in the mid-1980s [8]. Industrial and conservation groups related to these three industries are important stakeholders that lobby the government to protect the lake. More recently, groups concerned with dust impacts on human health and snow retention have become a more vocal interest group [80,81]. These industries and conservation groups often have conflicting goals for lake management [82]. For example, diking for freshwater refuges and mineral extraction reduces the lake level and increases its salinity, reducing the productivity of brine shrimp. Nevertheless, the commercial and public interest in the lake creates a strong group of stakeholders who lobby for effective management.

In contrast to Great Salt Lake, the economic value of Lake Urmia has not been evaluated, but mineral exploitation and hunting are currently restricted because the lake is part of Iran's National Park system [83], thus, limiting potential income. Nevertheless, there are several local salt extraction units along the north shore and *Artemia* harvesting ponds at the river mouth of the southern rivers. A preliminary study on the feasibility of mineral extraction from Lake Urmia reported that the annual production of sulfate of potash fertilizer is technically possible and economically justifiable [84]. While agriculture is the most water-consuming sector in the Lake Urmia basin, it accounts for just 13% of regional gross domestic product and 28% of employment. In contrast, the service sector, having 1% of total water consumption, comprises the highest share of the gross domestic product (55%) and employment (44%) in the three provinces in the Lake Urmia basin (based on 2019 data from the Statistical Center of Iran).

Beyond its economic value, Iran's largest lake is venerated as a cultural resource, and swimming and boating were once important recreational pursuits and a source of

income for local communities [14]. The lake's desiccation has caused losses of more than US$ 1.6 million (2019) to ecotourism, cultural heritage, and recreational and educational activities [18,85]. Similar to Great Salt Lake, health and agricultural concerns related to salt dust emissions from the dry Urmia lakebed are a major concern of residents in the region [17,84,86].

*3.3. Management and Scientific Approaches Utilized*

Management of Great Salt Lake has largely been driven by ecologists and limnologists interested in the lake itself, whereas the management of Lake Urmia has been led primarily by engineers and hydrologists interested in water management. Public interest and the proximity of major universities and research centers to Great Salt Lake led to earlier and more publications on Great Salt Lake than for Lake Urmia, where many of the research institutions are located 500 km away in Tehran. A SCOPUS search for "Great Salt Lake" yielded 386 peer-reviewed publications stretching from 1889 to 2022 (Figure A2). In contrast, for "Lake Urmia" or "Lake Uromiyeh", we found only 184 publications not beginning until 1986. The proportion of publications on the limnology, birds, and ecology of the lake itself was nearly seven times greater for GSL than for Lake Urmia. In contrast, the proportion of meteorological and hydrological publications was 1.2 times greater for Lake Urmia than for Great Salt Lake. Additionally, Lake Urmia articles were two times more likely to have utilized remote sensing techniques than those for Great Salt Lake, indicative of the distance of the lake from universities in Tehran (Figure A2).

The recovery effort for Lake Urmia has focused primarily on water resource management in the basin with little effort to understand how this would influence the lake ecosystem. For example, the target recovery elevation of 1274.1 m and river discharge to the lake of 3.09 $km^3 \cdot yr^{-1}$ were arrived at by assuming this would provide an appropriate salinity (240 $g \cdot L^{-1}$ NaCl or 270 $g \cdot L^{-1}$ total salinity) for *Artemia* [87]. However, this target salinity was based primarily on an anecdotal observation of *Artemia* in one area of the lake where the salinity was 250 $g \cdot L^{-1}$ and their absence at 280 $g \cdot L^{-1}$ [88], without regard for whether 280 $g \cdot L^{-1}$ would provide an adequate salinity for *Artemia* to produce sufficiently to support birds. An additional problem with the analysis was that a lake evaporation rate of 960 $mm \cdot yr^{-1}$ was assumed to be accurate [89], but in reality, this difficult-to-measure parameter has been estimated to range from 580–2000 $mm \cdot yr^{-1}$ (n = 16) with an average of 1320 $mm \cdot yr^{-1}$ [17]. If the mean evaporation estimate of 1320 $mm^{-1}$ were correct, then the water required to reach the ecological level has been underestimated by 38%. Furthermore, uncertainty and dynamic changes in Lake Urmia's bathymetry [14] together with the hysteresis between the lake volume and salinity [90], complicate reaching the target elevation and salinity. Consequently, additional research on the lake is clearly needed to address the underlying assumptions of the current target lake level of ULRP and arrive at a more reliable estimate for recovery.

The management of Great Salt Lake is primarily controlled by State agencies with the Utah Division of Forestry, Fire, and State Lands (FFSL) primarily responsible. However, the Utah Departments of Wildlife Resources (DWR), Water Quality (DWQ), and Geological Survey (UGS) provide most of the biological and chemical monitoring at many stations in the lake at bi-weekly (DWR) or quarterly (DWQ; UGS) intervals. These agencies, however, do not monitor or control the amount of water flowing into the lake. Those respective responsibilities are given to the U.S. Geological Survey and the Utah Division of Water Resources.

The FFSL also oversees the Great Salt Lake Advisory Council which is composed of representatives from the relevant management agencies, from the minerals and brine shrimp extractive industries, from county and municipal governments, and from environmental groups. This broad group of stakeholders often have competing objectives for managing the lake, but the Council provides a useful forum for working through complex management problems. A conservation group, FRIENDS of Great Salt Lake, also convenes at a bi-annual conference where all of the stakeholders can discuss issues. These multi-

disciplinary groups serve a major function of educating the public about the value of the lake and the problems it faces. Visitor centers at two State Parks and one of the diked wetland bird refuges also help in this education effort and promote public support for the lake.

A management plan for Great Salt Lake is revised at 10-year intervals, focusing on the different beneficial uses of the ecosystem [82]. The 2013 Plan resulted in a matrix showing the degree to which different uses are met at varying lake elevations (Figures 7 and A3). Many lake uses are supported within a 2.1-m elevation range from 1279.6 m (4198 ft.) to 1281.7 m (4205 ft.), but other uses are optimal at lower or higher lake levels. During its recorded history, however, the lake has fluctuated 9 m in elevation (Figure 3) demonstrating the difficulty of managing terminal lakes that respond both to natural climatic variation and human impacts. Despite the growing interest in the lake, and the availability of the beneficial use matrix, there is only a vague target elevation level suggested in the State's Comprehensive Management Plan (1280 m; 4200 ft.) [82], and no published estimates of how much inflowing water would be needed to obtain various lake levels. With receding lake levels, a target elevation and the flow needed to obtain it, are badly needed to assist water managers and politicians in their efforts.

In 2021, when Great Salt Lake first reached a record low level and worldwide media attention was focused on the problem (e.g., The Guardian) [91], legislation was finally passed that will begin to facilitate restoration [92]. Nevertheless, little water has actually been dedicated to saving the lake and competition for water from the agricultural sector and the growing municipalities is still a threat to the long-term survival of Great Salt Lake. There is also a threat of additional water development in the lake's watershed that is shared with the states of Idaho and Wyoming and managed with the tri-state Bear River Compact [93]. When this compact was initiated in 1958 and revised in 1980, all of the flow from the lake's major tributary was allocated to these three states, with none dedicated to the lake itself. Utah is beginning additional development of the river and the other states could choose to divert more water for agriculture and other uses in the future. The multi-state threat to further deplete flows thus represents a difficult political issue for lake management, similar to the ongoing multi-state negotiations on the use of water from the U.S.'s Colorado River [94].

In the Lake Urmia Basin, there is also a tri-state compact between the provinces of East Azerbaijan, West Azerbaijan, and Kurdistan, to limit their water development and help maintain the ecological function of the lake. However, several organizations are primarily in charge of land and water governance, development, and conservation. While the Ministry of Agriculture (MA) aims at increasing crop yield and reaching self-sufficiency in strategic products, the Ministry of Energy is responsible for providing adequate water and energy for agriculture and other sectors. On the other hand, the Department of the Environment owns the Lake Urmia land and is in charge of the conservation of the lake and its dependent ecosystems. The lack of coordination between these agencies with different priorities is one of the key obstacles to the Lake Urmia restoration [17,94].The ULRP was formed as an umbrella organization, but it has not entirely resolved these conflicts. Ethnic diversity in the basin, lack of awareness among the farmers on advanced irrigation and cultivation techniques, high dependency of rural households on agricultural incomes from small farmlands, water allocation and distribution conflicts between different users, and low involvement of local stakeholders in critical policies and management decisions are additional socio-economic barriers for the restoration of Lake Urmia [95].

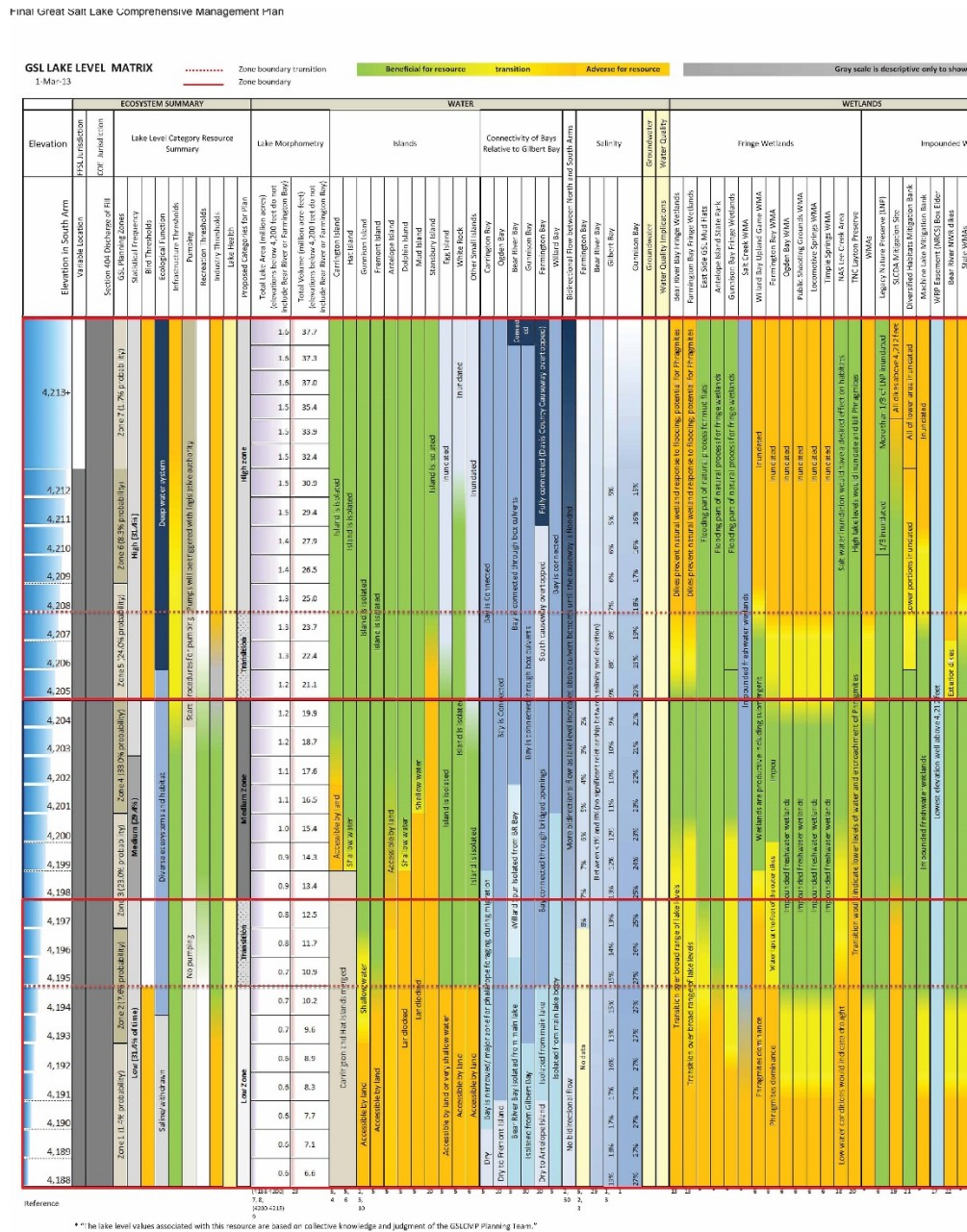

**Figure 7.** Lake elevation–Beneficial Use Matrix used by the Utah Division of Forestry, Fire and State Lands to help manage the lake. Ninety-three different uses are recognized, but only a portion are shown here (full matrix is available in Figure A3). Color-coding indicates how well these uses are supported at different lake elevations ranging from 1276.5 m (4188 ft) to 1284.1 m (4213+ ft).

In contrast to the management of Great Salt Lake, Lake Urmia has an approved water right. A total volume of 3.43 km$^3$·yr$^{-1}$ has been legally granted as the environmental water requirement of Lake Urmia under normal hydrological conditions. The ULRP had planned to supply this volume by implementing 27 restoration activities, particularly by: reducing agricultural water use by 40%; increasing environmental flow releases from reservoirs; discharge the treated effluents of Tabriz and Urmia wastewater treatment plants to the lake; and an inter-basin water transfer [96]. The transfer, originally planned for completion in 2019, will move 0.62 km$^3$·yr$^{-1}$ of water from the Lesser Zab River (a Tigris River tributary) through a US$ 3.2 billion, 36-km long tunnel [40,97]. However, the completion of the tunnel is now not expected until 2023. Consequently, after 9 years since the initiation of ULRP, no

water has been supplied from either the inter-basin transfer project or from the wastewater treatment plants because of financing and construction issues [97].

While agricultural water releases to Lake Urmia from upstream reservoirs of Lake Urmia have been reduced as a result of ULRP [97], it is not clear how much of the planned 40% reduction in agricultural water use has been achieved so far. Moreover, the saved water may not have reached Lake Urmia because of uncontrolled water withdrawals between dams and the lake. Records from downstream hydrometric stations (Iran Ministry of Energy) indicate that programmed environmental flows to Lake Urmia have been met in only 31% of the years since 1995. Even since ULRP began activities in 2013, Lake Urmia has only received its complete water allocation in 2 of 9 years. Overall, between 2013 and 2020 the lake has received 5.1 km$^3$ less water than was planned, resulting in a dysfunctional ecosystem.

One additional difference between the management of Great Salt Lake and Lake Urmia is the effort to preserve and enhance wetlands. At Great Salt Lake, wetlands are valued for bird habitat and recreational use, and agencies have allocated water and other resources to maintain them, sometimes at the expense of water reaching the main lake [3,50]. In contrast, the ULRP has focused solely on getting water to the lake itself to try to attain the ecological level of 1274 m [14], in part by river channelization that may limit environmental flows to several adjacent riverine wetlands. While environmental flows of Lake Urmia wetlands have been legally defined by the Department of Environment, data shows they have rarely been allocated in practice except in the wet years. Nonetheless, the value of Lake Urmia's nearby wetlands shouldn't be overlooked since they can serve as alternative habitats for birds, such as flamingos, that utilize wetlands, as well as open saline lakes [98,99].

One potential management option shared by both lakes is that they are pre-adapted for an "Aral Sea solution". That is, using diking to reduce the lake's surface area so that evaporation balances the diminished river inflows. Diking allowed 5% of the Aral Sea to continue functioning as the entire lake had before extensive water development desiccated the majority of the lake [6]. For both Great Salt Lake and Lake Urmia, the breaches in the existing transportation causeways could be closed or narrowed, allowing the southern portions of each lake to fill and function ecologically, but at the expense of all or some of the northern regions. In Great Salt Lake, closing the causeway would dry the north arm and connect Gunnison Island with the mainland, exposing one of the largest nesting colonies of White Pelicans (*Pelecanus erythrorhynchos*) in North America to predators, and their valuable habitat would likely be lost [100]. Additionally, the multi-million-dollar potash extraction industry there could not function. The present causeway dividing the lake, with only an 82-m breach connecting the two sides (Figure 8), already provides a partial Aral Sea solution, because without the causeway the entire lake would be too saline for *Artemia* and brine flies [31,101].

For Lake Urmia, sacrificing the northern portion of the lake would create dust hazards near the population centers of Urmia and Tabriz, and obviously, curtail recreation in that part of the lake. However, most of the islands harboring birds and mammals, as well as extensive wetlands are present in the southern part of Lake Urmia. Diking smaller sections of Lake Urmia or closing the causeway and filling the south arm at the expense of the north has been discussed for the lake restoration [40,102]. Greatly reducing the current 1250-m gap in Urmia's causeway would provide a partial solution similar to that of Great Salt Lake and would raise the level of the south arm, but still allow some water to flow to the north to provide dust control. However, sacrificing one area of each lake at the expense of another area is also fraught with political problems, particularly at Lake Urmia, where residents in the desiccated portion would bear the brunt of the negative effects (dust, lack of recreation, etc.). Nevertheless, as with the Aral Sea, saving a portion of each lake may be better than losing all of it. For both ecosystems, diking may need to be considered if climate change and population growth trends continue [103].

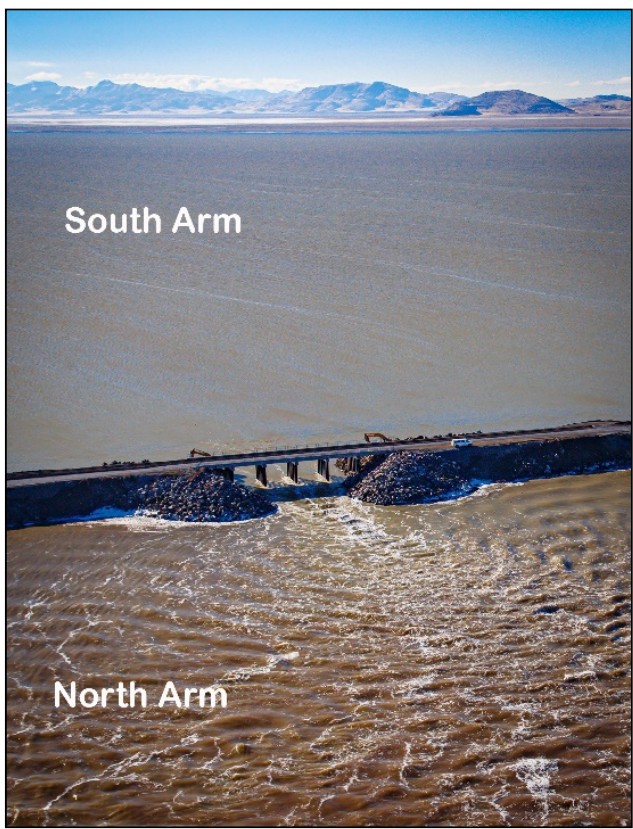

**Figure 8.** Flow of water through the 82-m breach in Great Salt Lake's railroad causeway shortly after the opening was completed. The narrow breach allows the South Arm, which receives all of the river inflow, to maintain a mean elevation 0.35 m higher than the north arm, and salinity over 120 g·L$^{-1}$ lower. Note the redish-brown water in the north arm due to an abundance of microbial Archaea.

### 3.4. Implications for Other Desiccating Endorheic Lakes

Many endorheic lakes have been entirely desiccated or are quickly losing water. Losses range from most of the 68,000 km$^2$ Aral Sea [104] to the far smaller Owen's Lake in California [105], both of which created economic hardship for surrounding communities, and continue to emit massive amounts of dust harmful to human health [104,106]. Some, like Mexico's Lake Texcoco (5400 km$^2$; [107]) and Nevada's Lake Winnemucca (300 km$^2$; [108]) were dried so long ago that they are almost forgotten. Other saline systems such as Kazakhstan's Lake Balkhash [109,110], Bolivia's Lake Poopó [111], and India's Sambhar Salt Lake [112] are in severe decline. Some, such as Kenya's massive, hyposaline Lake Turkana, are likely in severe decline [113], but too little information is available to confirm the magnitude of this change [114]. This lack of monitoring and other information highlights a problem in understanding threats to many saline lakes in developing countries [114], and for small saline lakes worldwide.

Throughout much of the developed world, cultural values are shifting, with environmental concerns becoming more prominent as the prosperity of the populace shifts from subsistence agriculture to more urban-based sources of income. In some cases, such as California's Mono Lake, this has led to the implementation of the *Public Trust Doctrine* by court decree to protect the ecosystems for wildlife and recreational uses at the expense of agriculture and urban purposes [115], while in others (e.g., Walker Lake, NV, USA) federal funds have been found to purchase water rights to save the lake [31].

As Williams [108] recognized 20 years ago, to maintain saline lakes worldwide, society and governments must recognize their economic, cultural, and ecological values and decrease water development in their basins. Failure to do so often results in economic hardship, health impacts on the populace, and the loss of biodiversity. To avoid these

problems our comparative analysis of the desiccation of Great Salt Lake and Lake Urmia suggest some useful considerations:

(1) A strong political will, financing mechanisms, and a transparent and collaborative involvement with local interest groups are required for successful preservation and restoration programs.

(2) For many systems, including Lake Urmia, restoration will require a shift from an agricultural-based economy to one based on manufacturing and services, as well as employment related to the lake's ecosystem services. This approach will tie the interest of the populace and ecosystem conservation together, instead of creating competition between consumptive uses of water and environmental uses.

(3) The diverse ecosystem services of saline lakes must be considered in project planning. These can include mineral extraction, recreation, bird habitats, and dust control, among others.

(4) Environmental uses of saline lakes normally include the surrounding, less saline wetlands. Consequently, managers must consider both the lake itself and these important surrounding ecosystems. These can often be maintained or restored with less water than is needed for the entire lake.

(5) Relying on expensive inter-basin water transfer projects for lake management is problematic: the costs and timing of these projects are usually grossly underestimated, and the donor watersheds are impacted by dewatering.

(6) It is far more difficult and sometimes more costly to recover water for a saline lake than to proactively plan for keeping water in a lake.

(7) Managers should closely monitor the restoration program in progress and learn from past experiences and adaptively update future activities.

**Supplementary Materials:** The following supporting information can be downloaded at: https://www.mdpi.com/article/10.3390/w14193005/s1.

**Author Contributions:** Conceptualization, W.A.W. and S.S.; methodology, W.A.W. and S.S.; formal analysis, W.A.W. and S.S.; investigation, W.A.W. and S.S.; data curation, W.A.W. and S.S.; writing—original draft preparation, W.A.W.; writing—review and editing, S.S. and W.A.W.; visualization, W.A.W.; supervision, W.A.W.; funding acquisition, W.A.W. All authors have read and agreed to the published version of the manuscript.

**Funding:** Funding for both authors was partially provided by the Semnani Family Foundation.

**Institutional Review Board Statement:** Not applicable.

**Informed Consent Statement:** Not applicable.

**Data Availability Statement:** See Supplementary Materials.

**Acknowledgments:** The authors appreciate Ali Hajimoradi and Behdad Chehrenegar from the Urmia Lake Urmia Restoration Program and Tohid Aligholonia and Hojjat Jabbari from Lake Urmia Futurology Center for providing relevant Lake Urmia data. We thank Naser Agh for hosting a conference on the recovery of Lake Urmia, upon which a portion of this manuscript is based. Ali Chavoshian provided the photo of Lake Urmia in the graphical abstract. Craig Miller, David Tarboton, David Rosenberg, and Sarah Null provided data for the analyses and many useful insights concerning the management of Great Salt Lake.

**Conflicts of Interest:** The authors declare no conflict of interest.

## Appendix A

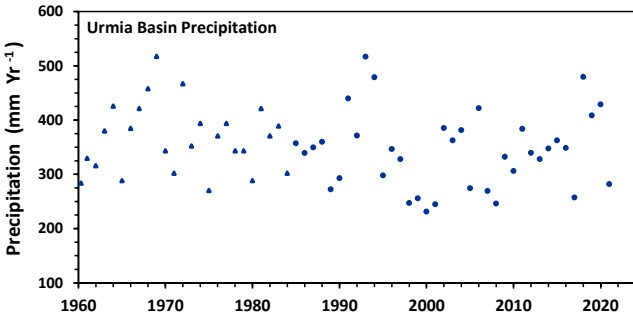

**Figure A1.** Precipitation in the Lake Urmia basin from 1960–2021. Since 1960, there has been no significant ($p$ = 0.27) decrease in precipitation in the Lake Urmia basin, although there has a slight negative trend of 0.56 mm·yr$^{-1}$. Data source: Triangles—Shadkam et al. 2016 [20]; Circles—Parsinijad et al. 2022 [17].

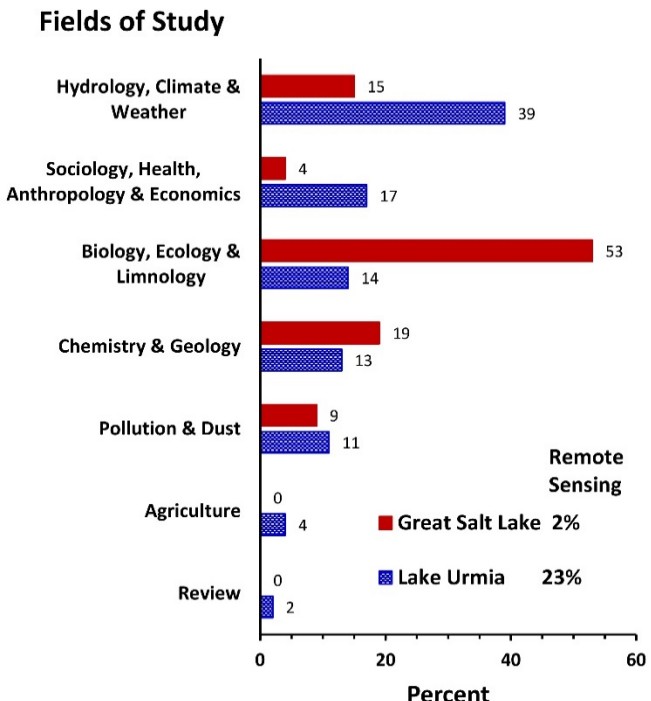

**Figure A2.** Percentages of research published in journals from 1889–2022 in different subject areas for Great Salt Lake and for Lake Urmia, and the percentage of studies that used remote sensing. SCOPUS data base. Note that the SCOPUS data base does not include papers published in Persian, which may have under-estimated earlier work on Lake Urmia.

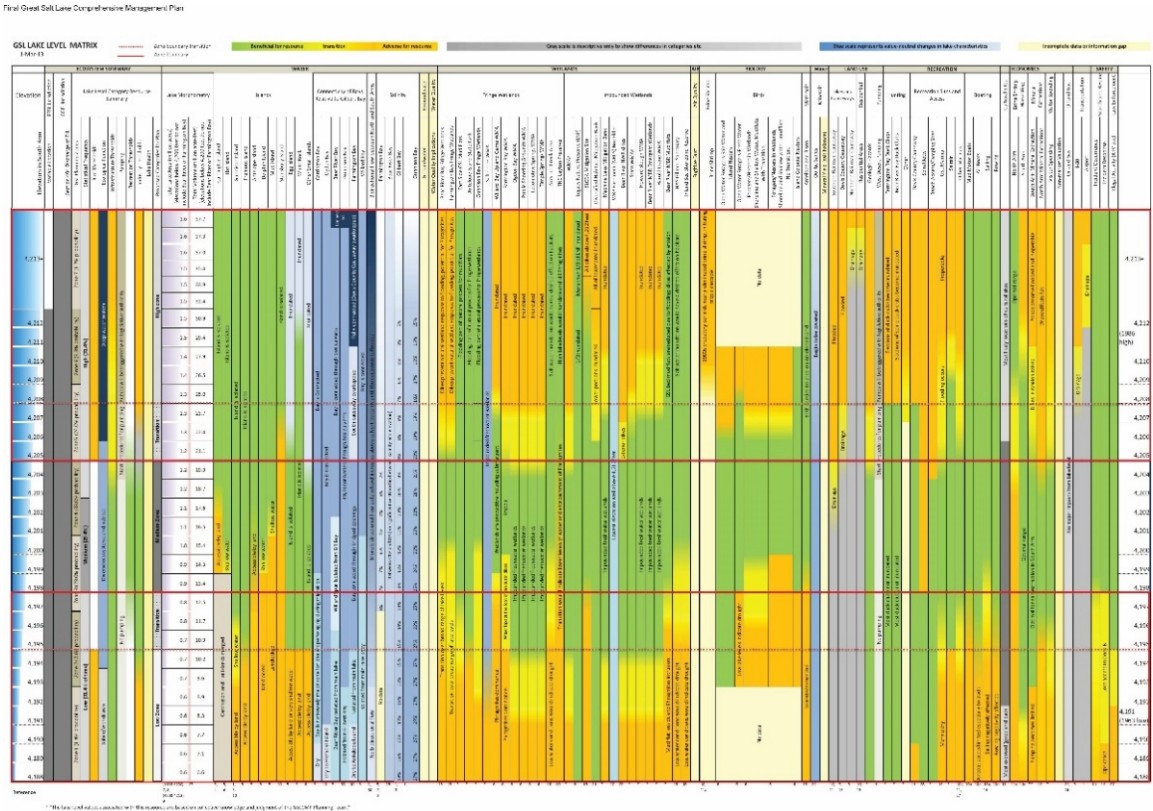

**Figure A3.** Lake elevation—Beneficial Use Matrix used by the Utah Division of Forestry, Fire and State Lands to help manage Great Salt Lake. Ninety-three different uses are recognized. Color-coding indicates how well these uses are supported at different lake elevations ranging from 1276.5 m (4188 ft) to 1284.1 m (4213+ ft).

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
