# Peer review of "Contrasting Management and Fates of Two Sister Lakes: Great Salt Lake (USA) and Lake Urmia (Iran)"

_water, doi:10.3390/w14193005_

Round 1
Reviewer 1 Report
The salinity and volume of both lakes were not adversely affected at the same rate. Please highlight this difference in the summary section.
Author Response
The reviewer highlighted an important point that water loss and salinity increase in Lake Urmia was far faster than that of Great Salt Lake. Consequently, we changed the Abstract to read:
These demographics led to a rapid increase in reservoir construction since 2000 and the subsequent loss of 87% of Lake Urmia’s volume. "The water development of Lake Urmia was later, but much faster than that of Great Salt Lake, causing Urmia’s salinity to increase from 190 to over 350 g L-1 in just 20 years, with subsequent severe ecological decline. "
Additionally, in Section 2.2 we added the following text to highlight this issue:
"Extensive water development in the Lake Urmia basin occurred later, but much more rapidly than in the Great Salt Lake basin, so the decline in Lake Urmia has been precipitous and more easily recognized (Figure 3, 4). "
Finally, we added the following short paragraph to the Discussion:
"
In many saline lake basins water development has been gradual, and thus the cause of declining lake levels has not been obvious (e.g. Great Salt Lake, Lake Abert[71], USA), particularly when climatic cycles raise and lower the lakes dramatically, making trends difficult to observe. Careful modeling was necessary to understand that water development was, indeed, the major factor desiccating these two lakes [3,71]. For Lake Urmia and the Aral Sea [72] the rapid declines in lake level coincided with major dam construction and irrigation projects, so that it was easy to attribute the lake’s demise to water development. Water balance models are thus critical for assessing reasons for the decline of any saline lake [3].
Finally, we added the following short paragraph to the Discussion:
"In many saline lake basins water development has been gradual, and thus the cause of declining lake levels has not been obvious (e.g. Great Salt Lake, Lake Abert[71], USA), particularly when climatic cycles raise and lower the lakes dramatically, making trends difficult to observe. Careful modeling was necessary to understand that water development was, indeed, the major factor desiccating these two lakes [3,71]. For Lake Urmia and the Aral Sea [72] the rapid declines in lake level coincided with major dam construction and irrigation projects, so that it was easy to attribute the lake’s demise to water development. Water balance models are thus critical for assessing reasons for the decline of any saline lake [3]."
Reviewer 2 Report
The manuscript contrasted management and fates of the two of the world’s largest saline lakes. The study helped to understand the different desiccation of the two similar lakes and give the instructive suggestions to prevent their shrinkages. The manuscript was well organized and written. My advice is to publish it after careful check for some spellings, such as “ my” in Line 413.
Author Response
The reviewer found one of our spelling errors and we've corrected it, as well as several others in the manuscript:
their comment: "My advice is to publish it after careful check for some spellings, such as “ my” in Line 413. "
"my" changed to "by"
Reviewer 3 Report
English language must be improved
There are some typing mistakes (ex., lines 174 and 175)
Maps in Fig 1. must be updated (especially, the Great Salt Lake map)
Please mention the aim of the study of the two lakes (not similarity or difference), but, what are the objectives for studying the two lakes together?
Author Response
1) We have reviewed the entire manuscript and made a number of grammatical/word changes to improve the English
2) We have corrected the Spelling/word duplication noted in lines 174-175, as well as others.
3) We updated Figure 1 to include both 'natural' lake levels, as well as the severely desiccated state of both as of the summer of 2022. This will help readers appreciate how bad the situation is for each lake.
We also updated the satellite image for Lake Urmia in the Graphical Abstract to show its worsening condition since we originally submitted the manuscript.
4) The reviewer asked us to "Please mention the aim of the study of the two lakes (not similarity or difference), but, what are the objectives for studying the two lakes together?
In the last sentence of our Introduction we addressed this, saying: Here we provide a comparative analysis of the desiccation of the two lakes that provides insights on management decisions that may help save them and that are relevant to saline lake management worldwide.
To emphasize this objective, we added a similar sentence to the Abstract: Here we provide a comparative analysis of the desiccation of these two lakes that provides insights on management decisions that may help save them and that are relevant to saline lake management worldwide.'